# Slab steepening and rapid mantle wedge replacement during back-arc rifting in the New Hebrides

Karsten M. Haase [1] ✉, Marcel Regelous[1], Christoph Beier [2] & Anthony A. P. Koppers [3]

The effects of the composition and angle of the subducting slab and mantle wedge flow on tectonic and magmatic processes in island arcs and associated back-arcs are poorly understood. Here we analyse the ages and compositions of submarine lavas from the flanks and the floor of the back-arc Futuna Trough some 50 km east of Tanna Island in the New Hebrides arc front. Whereas >2.5 Ma-old back-arc lavas formed from an enriched mantle source strongly metasomatized by a slab component, the younger lavas show less slab input into a depleted mantle wedge. The input of the slab component decreased over the past 2.5 million years while the enriched mantle was replaced by depleted peridotite. The change of Futuna Trough lava compositions indicates rapid (10 s of km/million years) replacement of the mantle wedge by corner flow and slab steepening due to rollback, causing extensional stress and back-arc rifting in the past 2.5 million years.

Subduction zones are the most seismically and volcanically active plate boundaries on Earth. Subduction of oceanic lithosphere can cause either compression or extension in the upper plate[1], possibly depending on the strength of the slab pull force[2] and angle of the subducting plate[3,4]. Thus, back-arc basins are known in both continental and oceanic environments, and in many cases the extension is associated with volcanism and the formation of new lithosphere[1,5]. The upper plate rifting processes at subduction zones, the evolution of magmatic activity, and the causes for the rifting of the upper plate are poorly known and several models exist. For example, the extension in the upper lithospheric plate may be due to either slab rollback, or to retreat of the upper plate, and steepening of the slab may be caused by gravitational forces, but the corner flow of the mantle above the slab may also affect the angle of subduction[4,5]. A statistical analysis of subduction zones revealed that back-arc extension occurs above steeply (>50°) dipping slabs, whereas back-arc compression is observed above shallow (<30°) dipping slabs[3]. The magmas forming in the mantle at the island arc and at some back-arc volcanoes show the influence of the subducting slab with an enrichment of fluid-mobile elements like Ba and Pb relative to immobile elements like Nb or Ce[6]. Variations of the fluid-immobile elements in many island arc and back-arc lavas suggest variable mantle sources ranging from depleted to highly enriched compositions[7,8]. The magmas of the island arc front typically form 105 to 130 km above the subducting slab[9]. In back-arc lavas the relative enrichment of fluid-mobile elements and the degree of source depletion is often related to the distance from the subduction zone[10]. Thus, the spatial and temporal change between the magma compositions in arc front and back-arc volcanoes yield insights into the processes of mantle flow and depletion, the addition of fluids and partial melts of basalts and sediments from the subducting slab to the mantle, and of partial melting of the mantle[6,8].

The New Hebrides Island Arc (NHIA) formed ~10 Ma ago above the eastward subducting Loyalty Basin of the Australian Plate (Fig. 1) and its rapid clockwise rotation with a present trench migration of ~9 cm/yr opened the North Fiji Basin[11–13]. The D'Entrecasteaux Ridge collided with the NHIA some 2 to 3 Ma ago[14,15] causing relatively slow subduction rates in this sector (27−43 mm/yr), contrasting with rates between 90 and 170 mm/yr observed north and south of the collision

[1]GeoZentrum Nordbayern, Friedrich-Alexander-Universität Erlangen-Nürnberg (FAU), Schlossgarten 5, 91054 Erlangen, Germany. [2]University of Helsinki, Department of Geosciences and Geography, Research Programme of Geology and Geophysics (GeoHel), Helsinki 00014, Finland. [3]College of Earth, Ocean and Atmospheric Sciences, Oregon State University, 104 CEOAS Admin Bldg, Corvallis, OR 97331–5503, USA. ✉e-mail: karsten.haase@fau.de

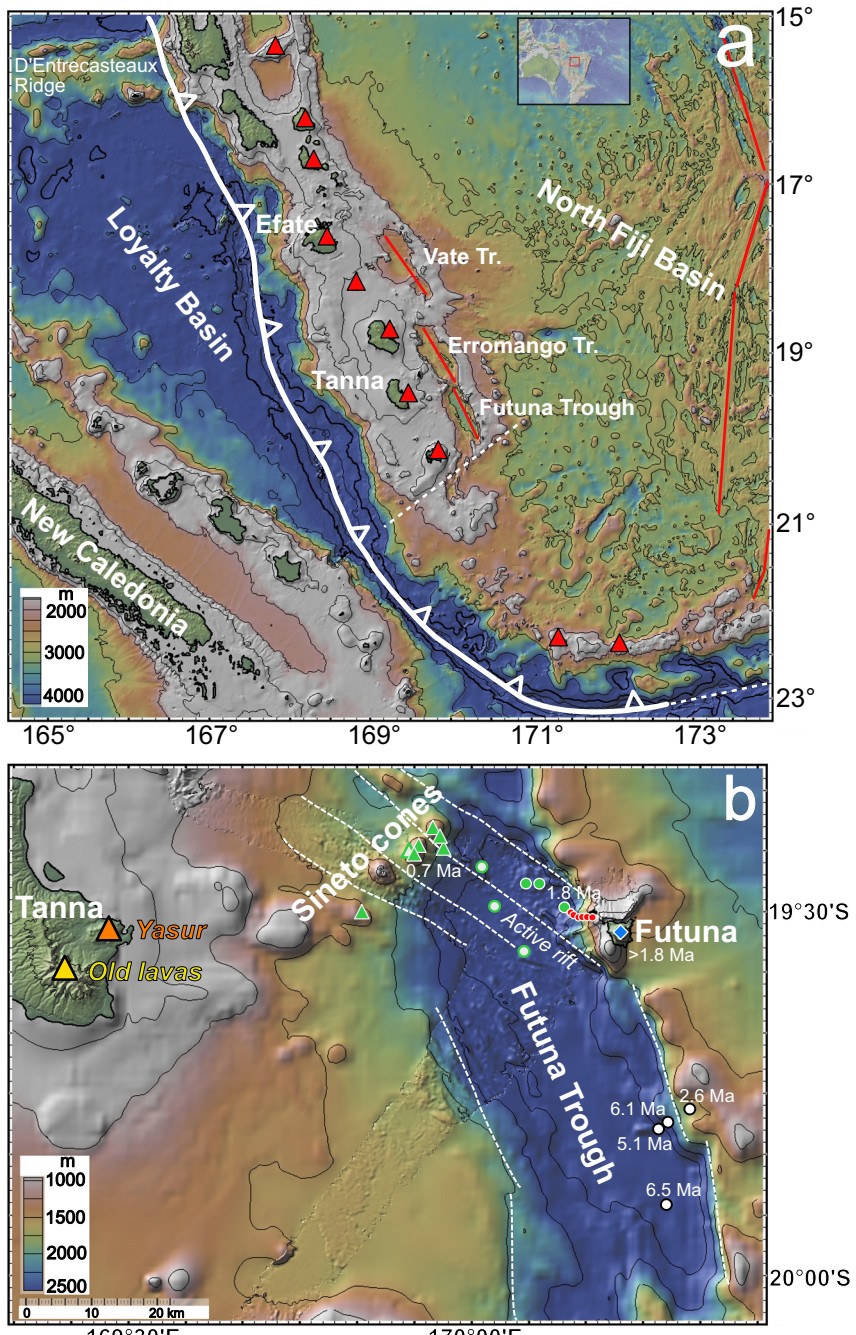

**Fig. 1 | Bathymetric maps of the study area in the southern New Hebrides island arc and in the northern Futuna Trough. a** Bathymetric map of the southern New Hebrides Island Arc (NHIA) with the red triangles showing the arc front volcanoes, and the Vate, Erromango, and Futuna back-arc troughs. Rift zones in the troughs and in the North Fiji Basin are shown as red lines. The outline of the New Hebrides subduction zone is indicated by the thick white line and fault zones by dashed white lines. **b** Bathymetric map of the northern end of Futuna Trough with the island of Tanna in the arc front with the location of the active Yasur volcano, the volcanoes of the Sineto cross-chain, the Futuna Trough and the island of Futuna. The different symbols show the location of samples discussed in this work as used in the following figures. Figure made with GeoMapApp (www.geomapapp.org), CC BY[69].

zone[16,17]. The NHIA segment south of the collision zone with the D'Entrecasteaux Ridge at 14°S to 17°S rotates clockwise (~9°/Ma) with respect to the North Fiji Basin causing rifting of the island arc and formation of the Vate, Erromango, and Futuna Troughs[18]. The crustal thickness beneath the New Hebrides island arc is 28 km, whereas the crust thins to ~10 km beneath the back-arc troughs[19]. In a global comparison of tectonic features of subduction zones, the New Hebrides subduction system is believed to be extreme in terms of rapid slab rollback as well as upper plate extension and advance[5], and

the steep landward trench wall and narrow forearc implies subduction erosion of the upper plate[20,21]. The Australian plate dips with ~67° below the southern New Hebrides island arc, which is one of the steepest slabs observed globally and the average depth between arc front volcanism and top of the slab H is $102 \pm 40$ km[9]. Heterogeneous Indian MORB-type mantle from the North Fiji Basin is believed to flow westwards beneath the NHIA, leading to abundant variations between depleted and enriched magmas in arc and back-arc volcanism[7,22,23].

The 25-30 km wide and 75 km long Futuna Trough (Fig. 1) is the southernmost back-arc basin (19°–20°S) and is bounded by NNW-trending normal faults[24,25]. The maximum depth is ~3600 m below sea level (mbsl)[26]. Futuna Island represents a horst structure[25] rising to 666 m above sea level east of the northern part of the trough, and has an area of 11 km². The slab lies at a depth of ~250 and 300 km beneath the Futuna Trough and Futuna island, respectively[27]. Foraminifera in tuffs indicate a Pliocene (5–3 Ma) age of the early submarine volcanism on Futuna[28], whereas the youngest known volcanism is represented by a hornblende basaltic andesite with a K-Ar age of 1.8 Ma[29]. The lavas are highly porphyritic and have a restricted compositional range from basalt to andesite, indicating significant fractional crystallization of melts derived from an incompatible element-enriched mantle[30,31]. The volcanic activity was followed by uplift of parts of the volcano above sea-level, erosion and later subsidence with the growth of coral reefs on top and on the flanks of Futuna Island[26]. Monjaret et al. [32] determined K-Ar ages of 2.6 to 6.5 Ma for calkalkaline lavas in the southern part of Futuna Trough (Fig. 1b), but these rocks probably represent the arc basement before back-arc rifting which started either 3 to 2.5 Ma[24,33] or 0.6 to 0.7 Ma[34] ago. Based on a positive magnetic anomaly observed in the Futuna Trough, Dubois et al. [25] suggested magmatic activity in the centre of the basins, possibly indicating the initial spreading of new crust. Side-scan sonar data of the Futuna Trough shows large areas of high acoustic reflectivity, but this was interpreted as being related to numerous faults rather than young lava flows[24]. Four volcanic edifices form a cross-chain at the northern end of Futuna Trough, where the southwestern-most structure of a cluster of three volcanic cones related to the young rift axis was named Sineto (Fig. 1b). The youngest lavas from Futuna Trough with an age of $0.7 \pm 0.2$ Ma[32] were recovered on the volcanic cone northeast of Sineto in the active rift (Fig. 1b). The island of Tanna lies at the arc front of the northern Futuna Trough (Fig. 1b) and consists of two Late Pliocene to Pleistocene volcanic units in the west of the island and a young late Pleistocene to Holocene volcanic unit, including the active Yasur volcano in the east[28]. Radiometric age data yielded K-Ar ages of 2.45 and 0.65 Ma for the older lavas and 0.23 Ma for the young unit[28,29].

Here, we show a systematic temporal compositional variation of the Futuna Trough back-arc magma compositions associated with the tectonic evolution of the rift. Our results indicate that the mantle in the back arc is replaced within a million years implying fast lateral mantle flow of several 10 s of km/million years in agreement with rapid trench-directed migration of the upper plate. The decreasing input from the slab during rift development indicates the steepening of the slab, which supports models of extension of the upper plate as a result of slab rollback. The fast trench-directed mantle flow may contribute to the slab steepening in addition to the gravitational forces.

## Results

### Sampling and age determinations of the Futuna Trough lavas

Thirty-five rock samples were collected between 3271 and 1008 mbsl from the base of and along the steep eastern wall of the Futuna Trough during two dives (39ROV and 44ROV) using the Remotely Operated Vehicle (ROV) Kiel 6000 (Fig. 1b) during expedition SO229 Vanuatu of the German Research Vessel R/V Sonne. Twenty-one of the recovered samples were used for this study (Supplementary Data 1) because the others were volcaniclastic rocks and/or not in situ. Additionally, a TV-guided grab (40TVG) and a wax corer (48VSR) recovered glassy basalt from the floor of Futuna Trough (Fig. 1b). Three volcanic cones on the northwestern rim of Futuna Trough were also sampled by TV grab and wax corer, including two volcanoes of the Sineto cross-chain (Fig. 1b).

The ⁴⁰Ar/³⁹Ar age dates of most samples were determined on plagioclase phenocrysts (Supplementary data 2, Supplementary Fig. 1) but groundmass was dated in sample 44ROV-23 and in the aphyric basalt 39ROV-01 from the floor of Futuna Trough. This sample contained fresh glass and very little radiogenic Ar yielding a plateau age of

$1.81 \pm 0.59$ Ma, whereas the total fusion age is $2.51 \pm 0.55$ Ma. The two ages are similar within the large uncertainty and an age of ~2 Ma is suggested for this sample. The ages obtained for lavas exposed along the profile of the eastern Futuna Trough rift flank are generally older. Plagioclase phenocrysts from lavas collected at different water depths yield plateau ages between 1.28 Ma and 3.44 Ma (Supplementary data 2, Supplementary Fig. 1) and agree with the Pliocene age of tuffs on Futuna Island[28] and the K/Ar age of $1.80 \pm 0.05$ Ma for a basaltic andesite[29]. The groundmass of sample 44ROV-23 yielded a more precise plateau age of $3.15 \pm 0.02$ Ma than the plagioclase plateau age of $3.50 \pm 0.50$ Ma. The new age data show that most lavas sampled along the Futuna Trough rift flank are >2.5 Ma similar to the old Futuna island lavas, whereas the rocks from the Futuna Trough floor are younger (< 2.5 Ma) and indicate volcanic activity in the back-arc basin.

### Chemical and isotopic composition of the Futuna Trough lavas

The recovered lava compositions range from 48 to 60 wt.% $SiO_2$, i.e. from basalt to andesite (Fig. 2) with the andesitic lavas occurring at the top of the profile at depths between 1099 and 1076 mbsl. Basalts from the central Futuna Trough floor generally have very low $K_2O$ contents <0.15 wt.%[35] and the Sineto cone lavas also largely belong to the low-K group (Fig. 2). In contrast, most lavas from the Futuna Trough flanks, Futuna island, and the Tanna volcanoes have medium-K compositions but particularly the young lavas from Yasur volcano on Tanna are high-K basalts and andesites (Fig. 2). The N-MORB normalized incompatible element patterns of the Futuna Trough floor basalts also show the lowest contents in fluid-mobile elements like Ba, U, and Pb as well as lower contents of the light REE (Fig. 3). In contrast, the samples from Futuna, the Futuna Trough flank, and the Tanna volcanoes show a fluid-mobile enrichment with Rb, Ba, and Pb up to 10-times higher than in N-MORB. Samples from the Sineto cones lie between the compositions of the Futuna Trough floor and Futuna/Tanna compositions (Fig. 3). Figure 4 shows the differences in the incompatible element enrichment between the Futuna Trough floor and Sineto cone lavas on the one hand, and the Futuna Trough rift flank and Futuna island lavas on the other hand. The relatively old Futuna Trough rift flank and Futuna island lavas have high Nb/Zr and $(La/Sm)_N$, the younger back-arc lavas are depleted with low Nb/Zr, Nb/La and $(La/Sm)_N$. The age determinations show the change of lava composition with time in the Futuna back-arc (Fig. 5). Most Futuna Trough flank samples, as well as Futuna island lavas have high Nb/La, Ba/Nb, and ²⁰⁶Pb/²⁰⁴Pb ratios, but low Ce/Pb <10 and ¹⁴³Nd/¹⁴⁴Nd (Figs. 5 and 6), whereas the young back-arc lavas from the Futuna Trough floor and Sineto cones have low ²⁰⁶Pb/²⁰⁴Pb, high ¹⁴³Nd/¹⁴⁴Nd, and Ce/Pb of 10 to 20[35] (Figs. 5 and 6). The lavas from the island arc front volcanoes on Tanna show relatively constant compositions over an age range of 2.5 Ma and have low Ce/Pb and Nb/La at intermediate Nd isotope ratios of ~0.51305 (Fig. 5).

In terms of incompatible element and isotope ratios we can again define two trends for the Futuna back-arc lavas (Fig. 6). One trend is formed by the old lavas from the Futuna Trough flank and Futuna island with ages >2.5 Ma and shows relatively constant high ²⁰⁶Pb/²⁰⁴Pb of ~18.65 and variable Nb/La and Ce/Pb as well as ²⁰⁸Pb/²⁰⁴Pb and ¹⁴³Nd/¹⁴⁴Nd ratios (Figs. 6 and 7). The compositions of these samples lie between those of the Tanna arc front lavas and the North Fiji Basin basalts with high ²⁰⁶Pb/²⁰⁴Pb, Nb/La and Ce/Pb ratios (Fig. 6). The second trend is defined by the lavas younger than 2.5 Ma from the Futuna Trough floor and the Sineto cones and shows decreasing ²⁰⁶Pb/²⁰⁴Pb with increasing Nb/La, Ce/Pb, and Ba/Nb (Fig. 6). These lavas have lower ²⁰⁸Pb/²⁰⁴Pb for a given ²⁰⁶Pb/²⁰⁴Pb than the older samples and the Tanna lavas and lie on a trend to North Fiji Basin basalts with lower ²⁰⁶Pb/²⁰⁴Pb. The young Futuna back-arc lavas show variable ²⁰⁶Pb/²⁰⁴Pb at relatively constant ¹⁴³Nd/¹⁴⁴Nd ratios and lie between the compositions of the arc front rocks from Tanna island and the incompatible element-depleted basalts from the North Fiji Basin (Figs. 6 and 7). Modelling suggests that >60% of the Pb and Nd in the

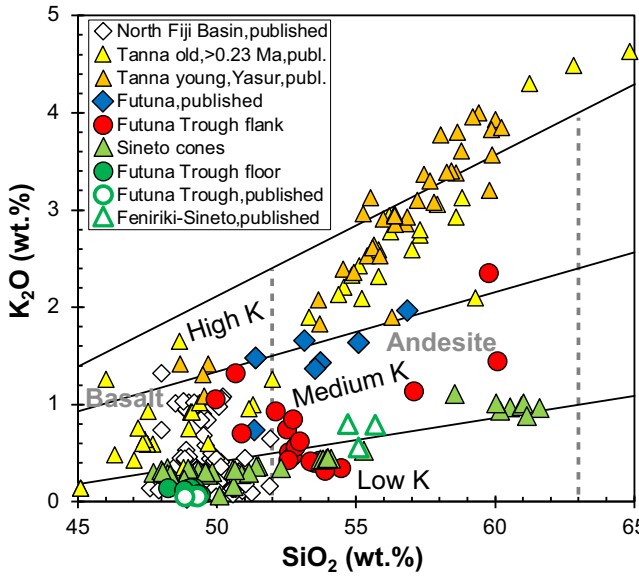

**Fig. 2 | Classification of the lavas based on K2O and SiO2 concentrations.** The K₂O contents versus SiO₂ diagram[70] showing the composition of the lavas from the old arc front volcanoes on Tanna and the young Tanna lavas from the active Yasur volcano[31,37,44,71–73], from the volcanoes of the Sineto cross-chain[32], the Futuna Trough, and the island of Futuna[30,35,44,74].

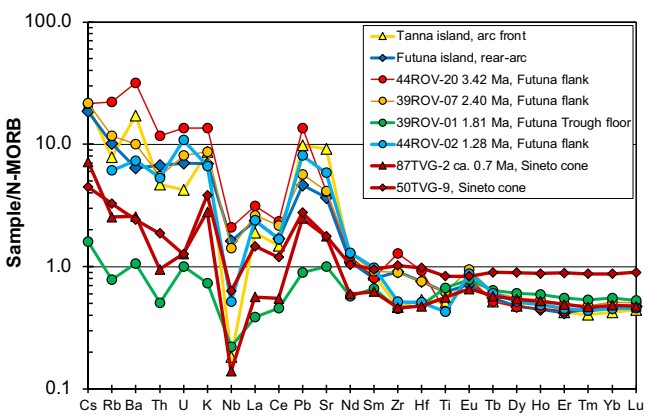

**Fig. 3 | N-MORB normalized incompatible element concentrations of representative lavas.** N-MORB[75] normalized diagram for lavas from Tanna, from the cross-arc volcanoes of the Sineto cross-chain (87TVG and 50TVG), the Futuna Trough floor (39ROV-01), the Futuna Trough flank (44ROV, 39ROV-07), and the island of Futuna[44].

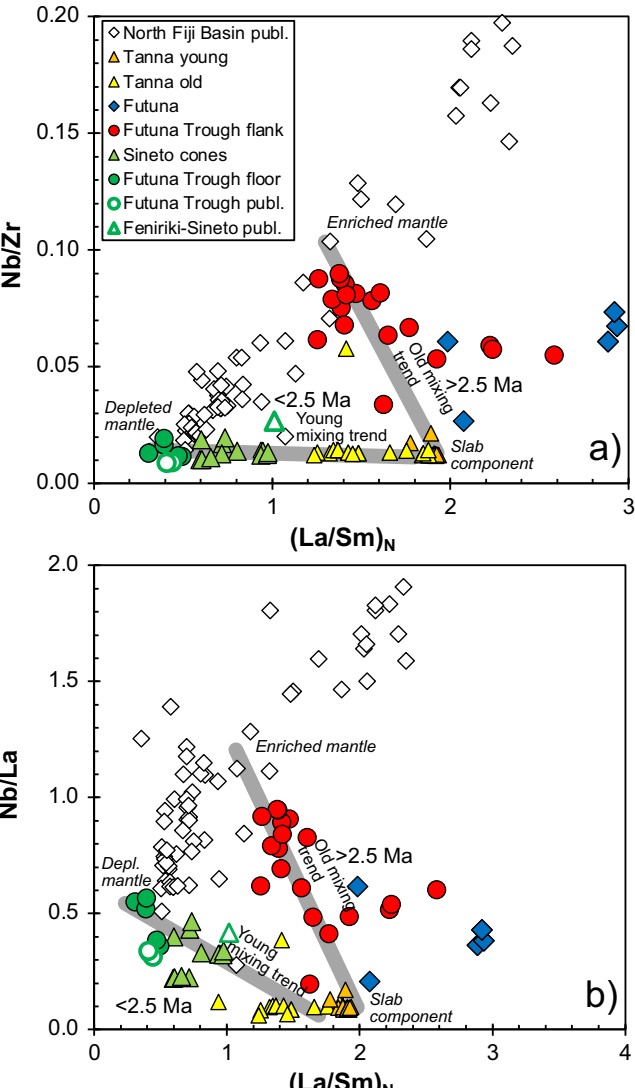

**Fig. 4 | Incompatible element ratios showing distinct lava sources.** Variation of **a** Nb/Zr versus (La/Sm)_N and **b** Nb/La versus (La/Sm)_N of lavas for the Tanna arc front and Futuna back-arc lavas. North Fiji Basin basalt compositions[46,76] are shown for comparison. Note that the Futuna Trough and island lavas have incompatible element-enriched compositions whereas those from the Futuna Trough floor and Sineto cross chain cones are depleted, i.e. we can define two groups of lavas along two mixing lines outlined in grey. Other data sources as in Fig. 2.

older lavas from Futuna and the Futuna Trough flank is derived from the slab, in contrast to <6% of Pb and Nd in the younger lavas (Fig. 7).

## Discussion

The uppermost (sampled at ~1000 mbsl) lavas from the northeastern flank of Futuna Trough and below Futuna Island (Fig. 1b) have ages of 3.42 and 3.15 Ma similar to those at the base of steep flank at 3170 mbsl and to the older lavas from Futuna Island. Thus, the about 2 km of lava of the rift flank formed rapidly within a few hundred thousand years before rifting of the trough and were exposed by normal faulting forming the steep cliff (Fig. 1b). The fact that lava samples with ages of 2.4 and 1.3 Ma occur between 2500 and 2000 mbsl on the rift flank is probably due to tectonic displacement along normal faults. Older lavas with calkalkaline composition[36] and ages of 2.6 to 6.5 Ma were sampled

at the southeastern flank of Futuna Trough (Fig. 1b) further south of Futuna Island[32], indicating that pre-rift magmatic activity in the back arc of the New Hebrides initiated >6 Ma ago[33]. The oldest lavas from Tanna and Futuna islands erupted >2.5 Ma ago[29] prior to opening of Futuna Trough, and the volcanoes were then rifted apart leading to the steep flanks of the islands[25,32]. The presently active volcano Yasur[37] occurs on the east coast of Tanna (Fig. 1b) and the eastward-directed migration of volcanic activity may be due to subduction erosion along the New Hebrides trench[20] similar to the migration observed in the Aleutian island arc[38]. Numerous volcanic structures occur in Futuna Trough east of Yasur volcano including the Sineto cones (Fig. 1b). The youngest of these volcanoes lies in the centre of the rift axis (Fig. 1b) and has a K/Ar age of 0.7 Ma (Dredge D20)[32]. Similar cross-arc volcanic chains between the arc front and the back-arc are known from other subduction zones like the Kermadec and Izu-Bonin island arcs[39,40]. The most MORB-like lavas occur as young glassy lava flows on the floor of the northern Futuna Trough (Fig. 1b) with an age of ~2 Ma implying that

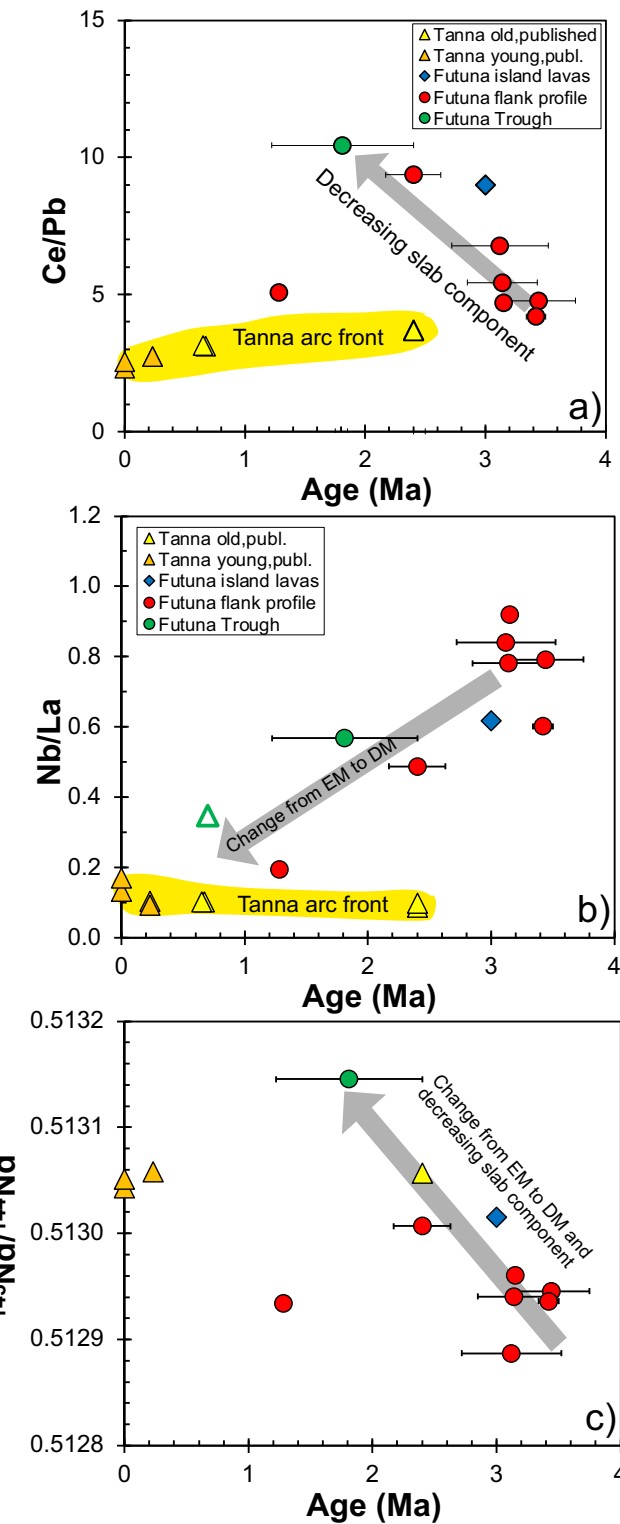

**Fig. 5 | Variation of incompatible element and 143Nd/144Nd ratios in the lavas with age.** Variation of **a** Ce/Pb, **b** Nb/La, and **c** $^{143}Nd/^{144}Nd$ with age (with 2σ error bars) for Tanna volcanoes and the Futuna back-arc lavas. Whereas the Tanna lavas show relatively constant compositions for the past 2.5 Ma, those of the back-arc vary considerably with time showing less slab component in the young lavas and a change from enriched to depleted mantle sources. Data sources as in Figs. 2 and 4.

thick crust beneath the NHIA[19], and these differences are comparable to those observed in other young back-arc basins like Bransfield Strait and the rifts in the Izu-Bonin island arc[41,42]. We conclude that volcanic activity occurred between 3.5 and 2.5 Ma prior to the rifting of Futuna Trough forming Futuna Island and the crust below the island that is now exposed on the rift flank. An initial rift probably existed when the northern Futuna Trough floor lavas erupted at ~2 Ma, but the presently active rifting is focused further to the west in a narrow rift axis with a width of 6.5 km (Fig. 1b). Concurrent with the rifting, volcanism also occurred on the flanks of the back-arc basin and particularly along the Sineto cross-arc volcanic chain which is probably younger than 2.5 Ma, with the youngest eruptions at 0.7 Ma[32] contemporaneous to MORB-like lavas on the Futuna Trough floor (Fig. 1b). Thus, both the island arc volcanoes on Tanna as well as the numerous volcanic structures in the back-arc display magmatic activity over a period of 3.5 Ma with migration of the back-arc volcanism towards the active rift zone of the Futuna Trough.

As shown in Figs. 4, 6, and 7 the lavas from the Futuna back-arc lie can be divided into two groups depending on their age and compositions. The lavas older than 2.5 Ma from Futuna Island and Trough flanks indicate mixing of an enriched mantle source resembling the most enriched basalts from the North Fiji Basin with a component from the subducting slab. The enriched mantle source causes the high Nb/Zr, Nb/La and low $^{143}Nd/^{144}Nd$ in some samples (Figs. 4, 6 and 7). The slab component is similar to that found in rocks from Tanna Island and causes the variation of Nb/Zr, Nb/La and $^{143}Nd/^{144}Nd$, as well as the low Ce/Pb and $^{208}Pb/^{204}Pb$. The relatively low $^{208}Pb/^{204}Pb$ for a given $^{206}Pb/^{204}Pb$ and the high $^{143}Nd/^{144}Nd$ of the slab component in the Tanna lavas (Fig. 7) probably reflects a hydrous fluid from subducted basalts of the Loyalty Basin[43,44]. In contrast, all lavas younger than 2.5 Ma from the Futuna Trough floor and the Sineto cones are depleted in Nb relative to La (Fig. 3) and lie on mixing trends between depleted lavas from the North Fiji Basin and Tanna lavas (Figs. 4, 6 and 7). Some of the Futuna Trough samples have very low Pb isotope ratios (Fig. 7a) and their high Ce/Pb and low Ba/Nb (Fig. 6) also indicate the absence of a slab component[23]. These lavas resemble the most depleted basalts of the North Fiji Basin. The Futuna back-arc lavas are thus forming initially from an enriched and later from a depleted mantle end-member, with additional variations of isotopic compositions being due to variable influx of a slab component (Figs. 6 and 7). If the enriched end-member would be within the depleted mantle, we expect variable mixtures due to variable partial melting like they are observed in the North Fiji Basin basalts[45,46]. Rather, the very different Nd and Pb isotopic compositions of the enriched and depleted end-members (Fig. 7b) and the trend of increasing Nd isotopes with time (Fig. 5c) imply that the depletion is not due to partial melting of one mantle source, but reflects different sources. We conclude that the lavas <2.5 Ma formed from a distinct depleted mantle source that was affected by the slab component also mixing into the Tanna magmas.

Importantly, the basalts from the lava flows on the Futuna Trough floor indicate a minor slab input as indicated by their MORB-like Ba, U, and Pb contents (Fig. 3) and high Ce/Pb (Fig. 6). In fact, modelling of Pb and Nd isotope variations of samples from the Futuna Trough and the Sineto cross-chain suggests <6% slab contribution to the lavas <2.5 Ma, whereas the older lavas (>2.5 Ma) from Futuna and the Futuna Trough flank require a massive (>60%) input of incompatible elements by the assumed slab component (Fig. 7). The slab input did not change in composition at least in the last 3.5 Ma because the old and young Tanna lavas have similar compositions and closely reflect the slab component in elements like La, Nd, and Pb (Figs. 6 and 7). The observed change in the back-arc lavas apparently did not affect the degree of partial melting considerably, because both young and old lavas have similar concentrations of slightly incompatible elements like the heavy rare earth elements (Fig. 3). The increasing Ce/Pb ratios between 3.5 and 1.8 Ma in the Futuna back-arc (Fig. 5a) implies a

by this time, the back-arc basin had formed. Thus, we suggest that back-arc rifting in Futuna Trough was initiated at 3.0 to 2.5 Ma in agreement with previous models[24,33]. The newly formed crust of Futuna Trough has a thickness of about 10 km compared to the 28 km-

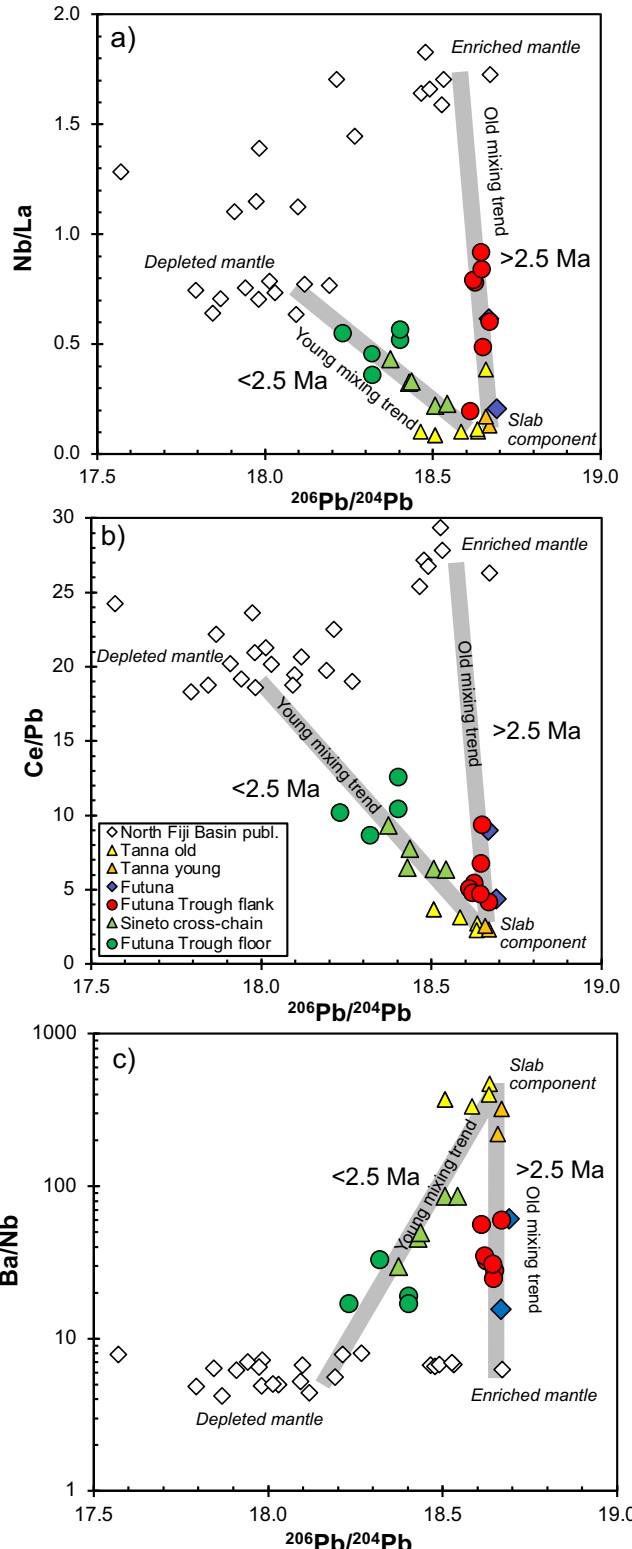

**Fig. 6 | Incompatible element ratios versus isotope ratios indicating mixing between different magma sources.** Variation of **a**) Nb/La, **b**) Ce/Pb, and **c**) Ba/Nb ratios with $^{206}Pb/^{204}Pb$ for the Tanna arc and Futuna back-arc lavas. North Fiji Basin basalt compositions[46] are shown for comparison. Similar to Fig. 4 we can define two mixing lines: 1) between enriched mantle and slab component, and 2) between depleted mantle and slab component. These two trends correspond to lavas with different ages, i.e. 1) lavas older than 2.5 Ma, and 2) to lavas younger than 2.5 Ma. Data sources as in Figs. 2 and 4.

decreasing input of the slab component with time. Typically, the slab input into back-arc magmas depends on the spatial arrangement of the slab relative to the back-arc volcanoes[10]. The release of the slab components occurs at specific depths due to the breakdown of hydrous minerals[47], but fluids or melts released at deep levels are probably carried deeper into the mantle with the solid flow[48]. Consequently, the width of the melting region in the arc and back-arc affected by the slab component strongly depend on the steepness of the slab[47]. At the

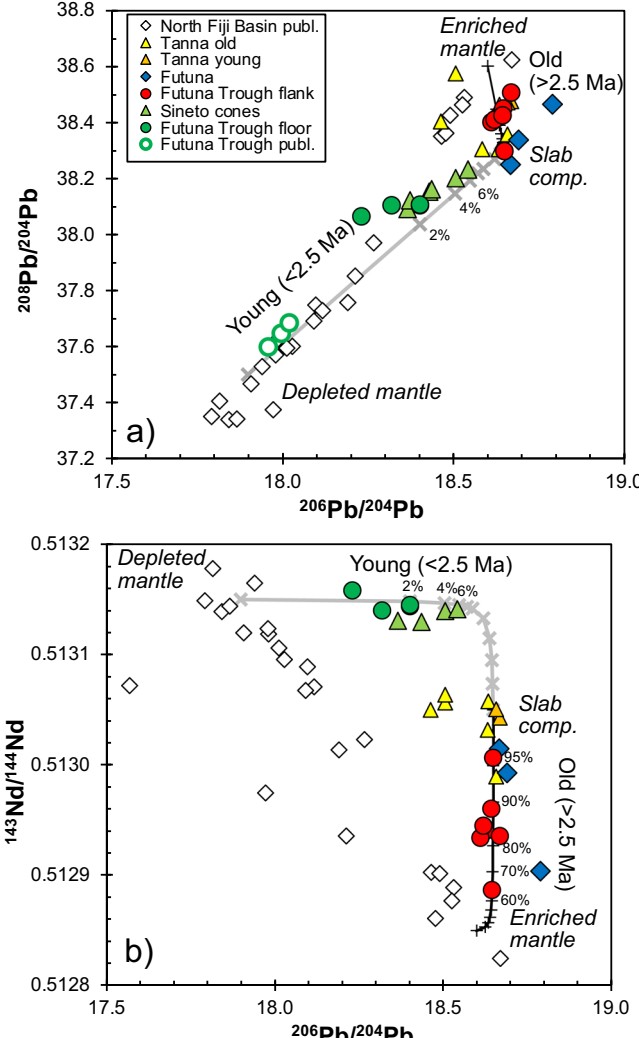

**Fig. 7 | Mixing of different magma sources indicated by radiogenic isotope ratios.** Variation of **a**) $^{208}Pb/^{204}Pb$ and **b**) $^{143}Nd/^{144}Nd$ versus $^{206}Pb/^{204}Pb$. The grey lines indicate binary mixing (1) between enriched North Fiji Basin mantle and the slab component, and (2) between depleted North Fiji Basin mantle and the slab component represented by Tanna lavas. The Futuna Trough floor basalts with the lowest Pb isotope compositions are from the centre of the northern basin[23] and may be the youngest lavas in Futuna Trough, see Fig. 1b. Numbers close to tick marks indicate percentage of addition of the slab component to the mantle. The depleted mantle end-member has $^{206}Pb/^{204}Pb$ of 17.9, $^{208}Pb/^{204}Pb$ of 37.55, and $^{143}Nd/^{144}Nd$ of 0.51315 with Pb and Nd concentrations of 0.014 ppm and 0.483 ppm similar to depleted MORB mantle[77]. The enriched North Fiji Basin MORB end-member has $^{206}Pb/^{204}Pb$ of 18.6, $^{208}Pb/^{204}Pb$ of 38.6, and $^{143}Nd/^{144}Nd$ of 0.51285 with Pb and Nd concentrations of 0.15 ppm and 2.5 ppm. For the slab component, we assume a $^{206}Pb/^{204}Pb$ of 18.65, $^{208}Pb/^{204}Pb$ of 38.3, and $^{143}Nd/^{144}Nd$ of 0.51305 with Pb and Nd concentrations of 1.4 ppm and 0.39 ppm estimated for a fluid from subducted basalt. Data sources as in Fig. 2, North Fiji Basin basalt compositions from Oh, et al.[46], and Pb isotope data for Futuna Trough samples from Heyworth, et al.[23].

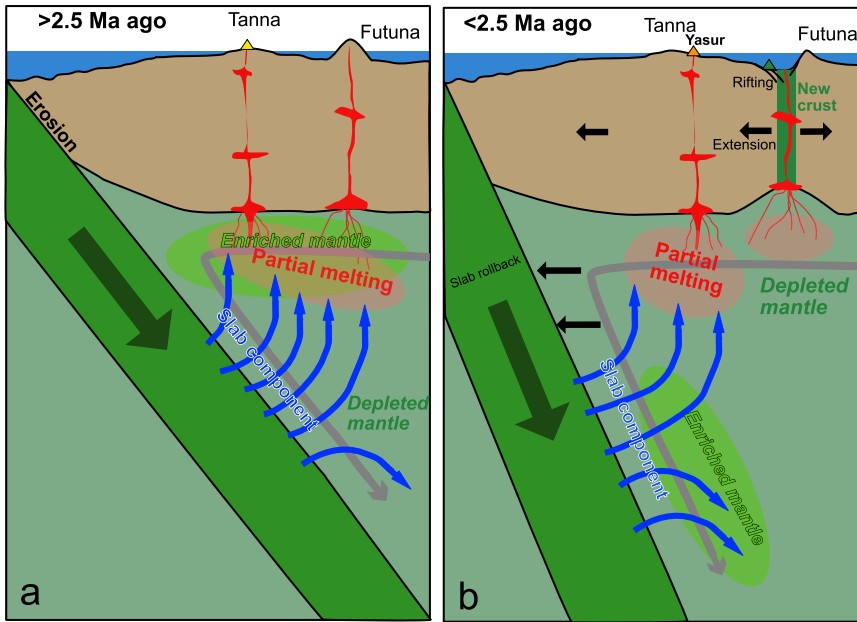

**Fig. 8 | Cartoon showing a model of the tectonic and magmatic processes.** The model shows the situation **a**) prior to 2.5 Ma ago and **b** less than 2.5 Ma ago. In **a** the slab dips less steeply than at present causing a wide effect of the slab component on mantle melting extending into the back-arc. In b) the slab steepens causing extension and rifting in the upper plate and narrowing of the influence of the slab component in the mantle wedge so that little slab effect is observed in the young lavas. See text for further explanation.

northern Futuna Trough, rifting and formation of new crust has not caused a large change in the location of back-arc volcanism relative to the trench, i.e. the Futuna Trough floor lavas and particularly the Sineto cross-chain lavas formed at a similar distance as the Futuna and Futuna flank lavas prior to rifting. Thus, we suggest that the decreasing slab input on the Futuna Trough magmas indicates a steepening of the slab some 2.5 Ma ago, so that younger magmas received less contribution from the slab. Today the Benioff Zone beneath the Futuna Trough lies at a depth of more than 250 km[27,49], which explains the small effect or even the absence of the slab component in the Futuna Trough lavas[23]. A similar pattern of narrowing of the volcanically active zone in the back-arc, migration of volcanism towards a back-arc rift zone close to the arc front, and decreasing slab input into the back-arc magmas is observed in rocks younger than ~8 Ma in the Izu-Bonin subduction zone[40,50]. At 2 to 3 Ma the back-arc rift lavas at the northern and central Izu-Bonin island arc show a significant decrease of the slab contribution to the mantle wedge, possibly from an addition of partial melts of subducted sediments prior to 2 Ma to hydrous fluids affecting the mantle after 2 Ma[50,51]. Steepening of the slab at the Izu-Bonin subduction zone was considered the most likely explanation for the observed changes in the occurrence and composition of volcanic rocks in the back-arc[50]. Consequently, we suggest that the similar migration of volcanic activity from a wide back-arc region towards a narrow back-arc rift close to the arc front as well as the decreasing slab input into the back-arc melting region in the southern New Hebrides and central to northern Izu-Bonin island arcs most likely reflect steepening of the slab.

The change from the enriched mantle source prior to 2.5 Ma to the depleted mantle source younger than 2.5 Ma indicates that the mantle forming the Futuna Trough magmas was replaced at ~2.5 Ma by depleted mantle typical of sources present beneath the North Fiji Basin[46,52]. As discussed above it is unlikely that this change was due to a depletion event in the mantle because it caused a sudden alteration of the incompatible element and isotope ratios, for example, from Nb/La from >1.0 to 0.7 and $^{143}Nd/^{144}Nd$ of 0.51285 to 0.51315 (Figs. 5a and 6b). The mantle source of both the old and young Tanna Island lavas resembles the depleted mantle of the younger lavas as indicated by the

similar Nb contents and Nb/Zr ratios of the Tanna lavas to those from the Futuna Trough and Sineto volcanoes (Figs. 3 and 4a). Consequently, the depleted mantle source may have also been present beneath the arc front at ~2.5 Ma.

Figure 8 shows a cartoon of the tectonic and magmatic processes during the formation of the Futuna Trough. The change from enriched to depleted mantle at ~2.5 Ma (Figs. 4, 5 and 6) indicates that the mantle wedge peridotite was rapidly replaced. A fast mantle flow is in agreement with the rapid advance of the upper plate with a velocity of ~60 km/million years suggested by Heuret and Lallemand[5]. In fact, seismic shear-wave splitting models indicate that the mantle flow in low-viscosity regions of the mantle wedge at subduction zones may be several times faster than the plate velocities[53], so that the rapid flow below Futuna Trough appears likely. Previously, the replacement of Pacific MORB-type mantle by Indian MORB-type mantle with velocities of 100 to 280 km/million years was suggested beneath Futuna Trough ~1.8 Ma ago, either by trench-parallel southward-directed or westward-directed mantle flow[23]. On the other hand, a northward-directed return flow around the slab edge at the southernmost New Hebrides Trench near 23°S was proposed[54], while modeling of the dynamic topography of the North Fiji Basin suggests hot shallow mantle upwelling affecting also the Futuna Trough region[55]. Thus, the direction of the mantle flow beneath Futuna Trough is poorly constrained, but given the fast rollback of the Australian plate[5] we suggest west-directed poloidal flow of North Fiji Basin depleted MORB mantle in agreement with previous geochemical models[23,35].

Prior to ~2.5 Ma the slab was probably less steep than after 2.5 Ma (Fig. 7) and caused back-arc compression as well as erosion of the upper plate which may have led to the eastward migration of the magmatic activity on Tanna. After 2.5 Ma, the slab input on the Futuna back-arc lavas decreased implying that the slab became steeper which caused the extension of the upper plate and the formation of normal faults (Fig. 7). Interestingly, the decreasing slab input into the Izu-Bonin back-arc magmas also occurred over a few million years and back-arc rifting is believed to have started at ~2.8 Ma in the central Izu-Bonin region[50,51]. Similarly, rifting of the Futuna Trough started at ~2.5 Ma and by 1.8 Ma the MORB-like magmas of the Futuna Trough

floor formed by adiabatic ascent of mantle beneath the rift, rather than by flux melting by hydrous fluids from the slab. Some 20 km of basaltic crust may have formed by this process which may explain the strong magnetic anomaly in the centre of the Futuna Trough[25]. The back-arc rifting at 2.5 Ma was probably caused by the collision of the D'Entrecasteaux Ridge with the New Hebrides island arc at 16°S that led to fast clockwise rotation by slab rollback of the southern New Hebrides island arc[56,57]. This onset of fast rollback may have caused the slab steepening at the southern New Hebrides island arc. Continuous extension but a stronger influence of slab material occurs at the boundary between the Futuna Trough and the Erromango Trough further to the north forming the small volcanic cones in the Sineto cross-chain (Fig. 1). This influence may be due to flow of slab material from beneath the active Yasur volcano to the east, resembling processes observed in the cross-chains of the Kermadec island arc[39]. A comparable change of the mantle source beneath a back-arc due to slab rollback was also observed in the Lau Basin where early Pacific asthenosphere was replaced by Indian MORB-type mantle[58]. MORB-like magmas occur in the back-arc regions of the Izu-Bonin, Mariana, Tonga-Kermadec, and South Sandwich island arcs[10,59–61], all of which show a steeply (> 30°) dipping slab in the upper 125 km like that of the New Hebrides island arc[3]. Consequently, the steep slabs may allow rapid flow in the mantle wedge and the replacement of metasomatized mantle by normal upper mantle material. The subduction zones with steep shallow slabs also typically show fast migration of the trench and upper plate[3] that may reflect rapid mantle flow. In addition, the relatively hot mantle occurring at the New Hebrides subduction zone[55] may also permit fast mantle replacement.

Our new geochemical and geochronological data allow new insights into the early formation of a back-arc rift and the processes occurring in the mantle wedge. We can show that slab rollback and steepening of the subducting plate affects the magma compositions in the back-arc and causes the extension in the upper plate as suggested by early geophysical models[1]. The change of the back-arc mantle source from enriched to depleted composition within less than 1 Ma indicates a rapid horizontal flow of mantle at several tens of kilometres/million years towards the subducting plate that may contribute to slab steepening[5] in addition to the gravitational forces. This flow led to replacement of the enriched mantle portion in the melting region beneath the Futuna Trough. The occurrence of MORB-like mantle with little slab input appears to be common in back-arc magmas in the New Hebrides, Izu-Bonin, Mariana, Tonga-Kermadec, and South Sandwich subduction zones which have steep (> 30°) shallow slabs[3] and which may allow the rapid replacement of metasomatized mantle wedge material.

## Methods

Volcanic glasses from two lavas (39-ROV-01 and 44-ROV-19) were separated and reduced to mm-sized chips, handpicked under binocular microscope, embedded in epoxy and polished for determination of major elements, $SO_3$ and Cl contents. Fresh cores of whole-rock samples were washed with deionized water, dried at 40 °C, coarse crushed and ground to powder in a vibratory agate disc mill. Glass beads were fused from the powders and were used to analyse the concentrations of major element oxides and selected trace elements on a Spectro Xepos Plus XRF spectrometer at the GeoZentrum Nordbayern. Loss on ignition (LOI) was determined by weighing ~1 g of sample before and after drying in a muffle furnace at 1050 °C for 12 h. Precision and accuracy were checked (Supplementary data 1) by analyses of the basalt standards BE-N ($n = 3$) and BR ($n = 2$) and are better than 2.5% for all major elements except P (9%). Trace element measurements were carried out using a Thermo Scientific X-Series 2 Quadrupole Inductively Coupled Plasma Mass Spectrometer (ICP-MS) at the GeoZentrum Nordbayern following the methods described by Lima et al.[7]. Sample solutions (dilution factor of 4000) were introduced into the plasma via an Aridus desolvating nebuliser to minimise molecular interferences, and mixed online with a Be, In, Rh and Bi internal standard solution in order to correct for instrument drift. Reproducibility and accuracy were monitored by periodic analyses of BHVO-2 standard (Supplementary data 1) and is generally better than 5%. Accuracy relative to GEOREM preferred concentrations for BHVO-2 is within 3%, except for Cs (10%).

Sample preparation and mass spectrometry for Sr-Nd–Pb followed the procedures outlined previously[7]. Isotope ratios were determined by thermal ionization mass spectrometry (TIMS) on a Thermo Scientific TRITON Plus in static multi-collection. Within run mass bias correction used $^{87}Sr/^{86}Sr = 0.1194$ and $^{143}Nd/^{144}Nd = 0.7219$ while a $^{204}Pb$-$^{207}Pb$ double spike (DS) technique was applied for Pb. All errors refer to reproducibility at 2σ standard deviation of the mean (2 SD), whereas 2σ within run errors shown in Supplementary data 2 are 2σ / √n − 1 (n = numbers of scans passing the outlier test). DS corrected NBS981 ($n = 23$) values are $^{206}Pb/^{204}Pb = 16.9412 \pm 0.0013$, $^{207}Pb/^{204}Pb = 15.4983 \pm 0.0014$, and $^{208}Pb/^{204}Pb = 36.7209 \pm 0.0035$. Sample data are reported relative to $^{87}Sr/^{86}Sr = 0.710250 \pm 0.000006$ for NBS987 ($n = 21$), $^{143}Nd/^{144}Nd = 0.511850 \pm 0.000007$ for the La Jolla standard ($n = 73$) (Supplementary data 2). Total chemistry blanks are below 100, 50, and 30 pg for Sr, Nd, and Pb, respectively, and thus considered negligible.

Eight lava samples from the profile of the Futuna Trough rift flank were selected for $^{40}Ar/^{39}Ar$ age dating at Oregon State University (OSU), USA, following the method outlined in Koppers et al.[62] (Supplementary data 2, Supplementary Fig. 1). The clean sample was crushed using a steel plated jaw crusher and the grain size fraction between 150-300 μm sieved and washed. The fraction was acid-leached with 1 M HCl, then 6 M HCl, 1 M $HNO_3$, 3 M $HNO_3$ and ultra-pure deionized water (all for about 60 min) in an ultrasonic bath heated to ~50 °C. The leached sample was irradiated for 6 h in the TRIGA nuclear reactor at OSU, together with the FCT NM sanidine flux monitor[63]. The individual J-values for each samples were calculated by parabolic extrapolation of the measured flux gradient against irradiation height and typically give 0.1-0.2% uncertainties (1σ). The $^{40}Ar/^{39}Ar$ incremental heating age was determined with a multi-collector ARGUS-VI mass spectrometer. After loading the irradiated sample into Cu-planchettes in an ultra-high vacuum sample chamber, it was incrementally heated by scanning a defocused $CO_2$ laser beam in preset patterns across the sample, in order to release the Ar evenly. After heating, the reactive gases were cleaned using a SAES Zr-al ST101 getter operated at 400 °C, and two SAES Fe-V-Zr ST172 getters operated at 200 °C and room temperature, respectively. Blank intensities were measured every three incremental heating steps for groundmass. For calculating the ages, the corrected decay constant of Steiger and Jäger[64] was used: $5.530 \pm 0.097 \times 10^{-10}$ yr$^{-1}$ (2σ)[65]. Incremental heating plateau ages and isochron ages were calculated as weighted means with $1/\sigma^2$ as weighting factor[66] and as YORK2 least-square fits with correlated errors[67] using the ArArCALC v2.7.0 software[68] available from the http://earthref.org/ArArCALC/ website.

## Data availability

All geochemical and geochronological data generated in this study are provided in the Supplementary data 1 and 2 that are available at https://doi.org/10.6084/m9.figshare.23924721.

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

## Acknowledgements

We gratefully acknowledge the help of Captain D. Korte, his crew of RV Sonne, and the ROV Kiel 6000 crew during sample recovery and the entire SO229 scientific party for a successful and pleasant cruise. We thank the Government of Vanuatu for permission for the cruise, R. Arculus, T. Tevi, H. Cook, and T. McConachy for their help in organizing the cruise, and S. Lima and F. Hauff for help with sample analysis. The comments by H. Gamaleldien helped to improve the quality of this work. We acknowledge the Bundesministerium für Bildung und Forschung (BMBF) for funding the cruise with grant 03 G 0229 to KMH.

## Author contributions

K.M.H. designed the study and interpreted the data, C.B. interpreted data, M.R. analysed the samples and interpreted the data, A.A.P.K. dated the samples.

## Funding

## Competing interests

The authors declare no competing interests.
