## [Peer Review file · Nature Communications]

REVIEWER COMMENTS

Reviewer #1 (Remarks to the Author):

This MS is very interesting investigating the early rifting process and the associated mantle flow pattern during back arc basin spreading by geochemical data. The logic of the story is quite clear (although the geodynamic evolution needs further demonstration), and the supporting evidence is robust. My overall suggestion is moderate revision. Below are some comments/suggestions that may help improving the MS.

1. I wonder if there is any other evidence supporting the slab steepening process. In the Izu-Bonin-Mariana subduction zone, slab steepening was suggested by the gradual narrowing area of the arc magma sum (Ishizuka et al., 2009) and the migration of the rear-arc volcanism towards the trench (Miyazaki et al., 2020). The more slab melt component found in the rear-arc basalts of the Izu-Bonin arc also reveals a deep mantle condition at which the temperature is high enough to induce slab melt (Miyazaki et al., 2020). I think it is necessary to find more evidence supporting your argument that the New Hebrides slab may have steepened since 2.5 Ma.

2. Slab-deepening process in the Izu-Bonin subduction zone leads to a series of rear-arc volcanisms featured by increasingly enriched features with La/Sm (Miyazaki et al., 2020). This is contrary to the arc geochemistry variation shown in this study which is characterized by the increase in depletion. I think a comparison could be made between the two subduction zones to find the key process controlling the differences, which may better explain your ideas.

3. The balance between slab rollback and slab steepening. The New Hebrides subduction is featured by fast slab rollback. In my opinion, fast slab rollback promotes slab flattening, rather than steepening.

4. The New Hebrides subduction evolved for quite a while (>10 Myr), and the subducted slab is quite deep, reaching the mantle transition zone. Why slab steepening occurred at this particular moment (i.e., 2.5 Ma)?

5. Further discussion is needed on the mantle flow pattern and the depleted mantle source

region, especially based on previous geodynamic modeling studies. Both poloidal flow and toroidal flow are intensive in this region (Schellart et al., 2007). Perhaps the authors can provide a 3D cartoon to show a more realistic tectonic and magma mode.

Some minor comments:

- Line 64, Fig. 1: The figure needs to be polished. The authors are suggested to use the same scale color bar in Fig. 1a and b. The location of Fig. 1b should be marked in Fig. 1a.
- There are many location names (a bit confusing). The authors know the area very well, but I do have a hard time to follow the names. Please label or draw the boundaries of the geological units as well as the geographic units in the figure explicitly (e.g., Futuna backarc, Futuna flank, Tanna volcanoes, Tanna arc front). What is Futuna back-arc (e.g., Line 196)? The Futuna island? Where is Tanna arc front?
- Lines 83-102: It's good to discuss the onset of the rifting, although it's debated. Could the authors provide the preferred onset time of the rifting (and the reasons) based on the previous studies?
- Line 172 Where is Fig. 8? Delete the extra "in".
- Figure 2: "Tanna young" samples have higher K₂O contents than that of the "Tanna old". What does this indicate?
- Line 185: It's good to mark the absolute age by linear color bar in Fig. 2. Same for the Figs. 4, 5 and 6.
- Line 188, Fig. 3: Yellow line is not for visualization in the plot.
- Line 196, Fig. 4b : What is the yellow zone?
- Line 229: I agree with the "6.5 Ma", but doubt about "2.6". What is the evidence that the pre-rift magma activity ended 2.6 Ma ago
- Line 257: The whole section is very interesting.
- Line 284: How to draw this qualified conclusion (e.g., <6% and >60%)?
- Lines 326-328: Why the decreased slab input imply slab steepening? Besides, should the upper plate extension be caused by slab rollback, rather than slab steepening?

References

Ishizuka, O., Yuasa, M., Taylor, R.N. & Sakamoto, I. 2009. Two contrasting magma types coexist

after the cessation of back-arc spreading. *Chemical Geology*, 266(3-4):274-296.

<http://dx.doi.org/10.1016/j.chemgeo.2009.06.014>.

Miyazaki, T., Gill, J.B., Hamelin, C., Debari, S.M., Sato, T., Tamura, Y., Kimura, J.I., Vaglarov, B.S., Chang, Q., Senda, R. & Haraguchi, S. 2020. The First 10 Million Years of Rear-Arc Magmas Following Backarc Basin Formation Behind the Izu Arc. *Geochemistry, Geophysics, Geosystems*, 21(10). <http://dx.doi.org/10.1029/2020gc009114>.

Schellart, W.P., Freeman, J., Stegman, D.R., Moresi, L. & May, D. 2007. Evolution and diversity of subduction zones controlled by slab width. *Nature*, 446(7133):308-311.

<http://dx.doi.org/10.1038/nature05615>.

Reviewer #2 (Remarks to the Author):

Review of the manuscript: "Geochemical evidence for slab steepening and rapid mantle wedge replacement during back-arc rifting in the New Hebrides island" By Haase et al.

I have enjoyed reading the manuscript, the authors used details geochemical, geochronological (Ar-Ar), and isotopic (Sr-Nd-Pb) data of basaltic rocks from Futuna trough/island to identify an interesting geochemical compositional change in the lava that formed before and after 2.5 Ma, that gives an indication about the change in mantle source from enriched to depleted ones and related these observations to slab rollback and opening of back-arc basin. After reading this MS, I can say that the authors did a very good job making the MS ready for publication after a few more suggestions.

There are some major comments for the authors to be considered:

1- Can you add a section about the petrographical description of the samples (+ images) in the supplementary section?

2- Ar-Ar dating figure 8 is missing!!!?

3- I didn't see any discussion about the Sr isotopes data!! Can you show the relationship between Sr vs. Nd and Pb isotopes?

Below are line (L) comments that hopefully are useful to the authors:

L38-41, 49-52: Add reference, please.

L44-48: Variation in FME can also be temperature-dependent. Gamaleldien et al. 2019 SR (subduction polarity paper) can add to references here.

L172: You jumped from Fig. 1 in previous sections to Fig. 8, which is missing I can't find it!!!.

L182: one sentence is needed to summarize this new age data.

L198: This is not clear, there is no difference between Nd values between old and young Futuna Flank data? And the same for Ce/Pb ratio and even the Tanna samples??

L272: Some other ratios Ba/Nb, U/Nb, Pb/Nd, can help in your discussion and show the differences in slab contributions between the two types, please see Gamaleldien et al, 2020, SR (start of subduction paper). I like to see these ratios vs. Pb, Nd, Sr isotopes. These figures can be added to supplementary.

Fig. 1: Please add a more general map to Fig. 1 to show the location for the general/public readers. The symbol for Tanna Yasur "red triangle" can't be found in the following figures!!!

Fig. 2: Can you add the age for Tanna's old and young lava ??

Fig 5: Can you add Ba/La vs Pb isotopic data?

Fig 6: Please explain how you did the mixing model?

Fig. 7: please label the different components of the figure such as oceanic crust, mantle wedge, I think this figure can be more visualized than the present one??

Reviewer #3 (Remarks to the Author):

This is an interesting, valuable paper. The different lines of evidence the authors use to build their story – rock ages, geochemistry, location, tectonic history – are effectively described and synthesized. The tectonic setting, strategy and conclusions are unique, so the paper is definitely appropriate for Nature Communications.

Nonetheless, the paper can be improved for greater clarity and impact.

Clarity - To be fully understood and effective, the paper needs some work. In a short work, readers will compare text and figures as they read, so the the figures should to be explicit and consistent.

1) The most important item is Fig. 1. The lettering in 1a is tiny (to my eyes). Every feature mentioned in the Results and especially Discussion (Evolution of the tectonic and magmatic processes) should be labelled so that readers can correlate the upper plate locations with interpretations of the sample chemistry (Geochemical evolution of the magmas) and processes occurring at depth (Implications of the geochemical evolution for tectonic processes). A box showing the location of 1b on Fig. 1a would be useful.

2) Also, in general, the figure captions are brief and should be expanded to cover everything on the diagram. Some examples – no explanation the different symbols on Fig. 1b, no explanation of the yellow field on Fig. 4, samples identified only as numbers on Fig. 3 legend (these are in the caption but are better on both), explanation of open green circles on Fig. 6.

3) Supplementary data: Some small editing needed – Supplementary Data 1 has all geochemical data for samples (Table S1) and standards (Standards Tab). Supplementary Data 2 has Ar dates and analytical parameters in Tabelle 1 and nothing in Tabelle 2.

Broadening the impact:

The authors emphasize that the mantle flow rate they infer is fast (line 55), so some comparisons with rates of mantle flow inferred for other arcs is needed. Confirmation of results with those of with Heuret and Lallemand is good, but comparison with mantle wedge flow rates estimated in the literature elsewhere would heighten the impact of the paper and enhance the value of the authors' conclusions.

The reviewer comments are in normal and our replies in bold text.

REVIEWER COMMENTS

Reviewer #1 (Remarks to the Author):

This MS is very interesting investigating the early rifting process and the associated mantle flow pattern during back arc basin spreading by geochemical data. The logic of the story is quite clear (although the geodynamic evolution needs further demonstration), and the supporting evidence is robust. My overall suggestion is moderate revision. Below are some comments/suggestions that may help improving the MS.

1. I wonder if there is any other evidence supporting the slab steepening process. In the Izu-Bonin-Marina subduction zone, slab steepening was suggested by the gradual narrowing area of the arc magmatism (Ishizuka et al., 2009) and the migration of the rear-arc volcanism towards the trench (Miyazaki et al., 2020). The more slab melt component found in the rear-arc basalts of the Izu-Bonin arc also reveals a deep mantle condition at which the temperature is high enough to induce slab melt (Miyazaki et al., 2020). I think it is necessary to find more evidence supporting your argument that the New Hebrides slab may have steepened since 2.5 Ma. **Presently, the New Hebrides slab is one of the steepest on Earth and the opening of the Coriolis Troughs indicates that the upper plate stress field became extensional some 2.5 Ma ago. We believe that the most likely explanation is the steepening of the plate which is supported by numerical and experimental models e.g. by Cagnioncle et al (2007, JGR) and Grove et al (2012, Nature).**

2. Slab-deepening process in the Izu-Bonin subduction zone leads to a series of rear-arc volcanisms featured by increasingly enriched features with time (Miyazaki et al., 2020). This is contrary to the arc geochemistry variation shown in this study which is characterized by the increase in depletion. I think a comparison could be made between the two subduction zones to find the key process controlling the differences, which may better explain your ideas. **We believe that the steep slab allows rapid inflow of unmetasomatized mantle which may not be the case in the Izu-Bonin. MORB-like mantle is also observed in back-arcs of other subduction zones with steep slabs like Tonga, South Sandwich and the Marianas and we have mentioned this in the discussion.**

3. The balance between slab rollback and slab steepening. The New Hebrides subduction is featured by fast slab rollback. In my opinion, fast slab rollback promotes slab flattening, rather than steepening. **Seismic data show that the slab at the New Hebrides is one of the steepest on Earth and it was shown that steep slabs correlate with fast rollback, e.g. by Lallemand et al 2005 and Heuret et al 2005.**

4. The New Hebrides subduction evolved for quite a while (>10 Myr), and the subducted slab

is quite deep, reaching the mantle transition zone. Why slab steepening occurred at this particular moment (i.e., 2.5 Ma)? **We believe that the slab rollback has probably occurred in several phases with slab steepening and the event at 2.5 Ma was only the last in a series of such events but this is of course highly speculative.**

5. Further discussion is needed on the mantle flow pattern and the depleted mantle source region, especially based on previous geodynamic modeling studies. Both poloidal flow and toroidal flow are intensive in this region (Schellart et al., 2007). Perhaps the authors can provide a 3D cartoon to show a more realistic tectonic and magmatic mode. **We think that Futuna Trough is too far distant from the slab edge and thus toroidal flow seems unlikely. Rather, poloidal flow from the east can explain the presence of North Fiji Basin mantle beneath Futuna Trough which is in agreement with previous suggestions.**

Some minor comments:

- Line 64, Fig. 1: The figure need to be polished. The authors are suggested to use the same scale color bar in Fig.1a and b. The location of Fig. 1b should be marked in Fig. 1a. **OK, Figure 1 was redrawn and we hope it is better now, showing all the locations used in the text.**

- There are many location names (a bit confusing). The authors know the area very well, but I do have a hard time to follow the names. Please label or draw the boundaries of the geological units as well as the geographic units in the figure explicitly (e.g., Futuna backarc, Futuna flank, Tanna volcanoes, Tana arc front). What is Futuna back-arc (e.g., Line 196)? The Futuna island? Where is Tanna arc front? **Figure 1 was redrawn and we tried to avoid too many location names in the text.**

- Lines 83-102: It's good to discuss the onset of the rifting, although it's debated. Could the authors provide the preferred onset time of the rifting (and the reasons) based on the previous studies? **The preferred onset is 3 to 2.5 Ma based on previous tectonic models which is now discussed in the text.**

- Line 172 Where is Fig. 8? Delete the extra "in". **Deleted, the figure is moved to the supplement.**

- Figure 2: "Tanna young" samples have higher K₂O contents than that of the "Tanna old". What does this indicate? **We have checked and the old and young Tanna lavas are similar in K₂O contents at a given SiO₂. The relatively constant compositions are also seen in Nb/La and Nd isotopes (Fig. 4). The comment maybe is related to the fact that the young Tanna lavas are generally more evolved and there are no basaltic lavas at Yasur. Thus, however, is probably related to the magma ascent which is not discussed in this work.**

- Line 185: It's good to mark the absolute age by linear color bar in Fig.2. Same for the Figs. 4, 5 and 6. **We are not sure whether we understand this comment but we have redrawn many of the figures..**
- Line 188, Fig. 3: Yellow line is not for visualization in the plot. **We changed the colour and think it is better visible now**
- Line 196, Fig. 4b : What is the yellow zone? **The Tanna arc front lavas, a label was added**
- Line 229: I agree with the "6.5 Ma", but doubt about "2.6". What is the evidence that the pre-rift magmatic activity ended 2.6 Ma ago. **We agree that the end of the magmatic activity is not clear and thus deleted this part of the sentence.**
- Line 257: The whole section is very interesting. **Thank you**
- Line 284: How to draw this qualified conclusion (e.g., <6% and >60>)? **The absolute amounts are of course dependent on the mixing model but the much larger input for the older lavas than for the younger should hold.**
- Lines 326-328: Why the decreased slab input imply slab steepening? Besides, should the upper plate extension be caused by slab rollback, rather than slab steepening? **Numerical models and experimental data suggest that the steeper slabs will effect a relatively narrow portion of the mantle wedge with fluids or melts (see, for example, Cagnioncle et al 2007 JGR and Grove et al 2012 Nature). Thus, the fact that we do not see the slab component in the back-arc after 2.5 Ma suggests to us that the slab steepened. We have explained this in more detail in the text and changed Figure 8. Statistic studies of extension in the upper plate suggest that steep slab roll back and thus cause extensional stress.**

References

- Ishizuka, O., Yuasa, M., Taylor, R.N. & Sakamoto, I. 2009. Two contrasting magmatic types coexist after the cessation of back-arc spreading. *Chemical Geology*, 266(3-4):274-296. <http://dx.doi.org/10.1016/j.chemgeo.2009.06.014>.
- Miyazaki, T., Gill, J.B., Hamelin, C., Debari, S.M., Sato, T., Tamura, Y., Kimura, J.I., Vaglarov, B.S., Chang, Q., Senda, R. & Haraguchi, S. 2020. The First 10 Million Years of Rear-Arc Magmas Following Backarc Basin Formation Behind the Izu Arc. *Geochemistry, Geophysics, Geosystems*, 21(10). <http://dx.doi.org/10.1029/2020gc009114>.
- Schellart, W.P., Freeman, J., Stegman, D.R., Moresi, L. & May, D. 2007. Evolution and diversity of subduction zones controlled by slab width. *Nature*, 446(7133):308-311. <http://dx.doi.org/10.1038/nature05615>.

Reviewer #2 (Remarks to the Author):

Review of the manuscript: "Geochemical evidence for slab steepening and rapid mantle wedge replacement during back-arc rifting in the New Hebrides island" By Haase et al.

I have enjoyed reading the manuscript, the authors used details geochemical, geochronological (Ar-Ar), and isotopic (Sr-Nd-Pb) data of basaltic rocks from Futuna trough/island to identify an interesting geochemical compositional change in the lava that formed before and after 2.5 Ma, that gives an indication about the change in mantle source from enriched to depleted ones and related these observations to slab rollback and opening of back-arc basin. After reading this MS, I can say that the authors did a very good job making the MS ready for publication after a few more suggestions.

There are some major comments for the authors to be considered:

1- Can you add a section about the petrographical description of the samples (+ images) in the supplementary section? **The petrography is not particularly important for this manuscript and thus we refrained from adding extensive petrography and pictures.**

2- Ar-Ar dating figure 8 is missing!!!? **Added as Figure S3**

3- I didn't see any discussion about the Sr isotopes data!! Can you show the relationship between Sr vs. Nd and Pb isotopes? **We are cautious about the Sr isotope data as some of the older samples and older data apparently are affected by seawater alteration and were not leached prior to analysis. For example, Sr isotope data for Tanna and Futuna lavas scatter considerably, whereas Nd and Pb isotope data give a more consistent trend.**

Below are line (L) comments that hopefully are useful to the authors:

L38-41, 49-52: Add reference, please. **Added**

L44-48: Variation in FME can also be temperature-dependent. Gamaleldien et al. 2019 SR (subduction polarity paper) can add to references here. **We agree, but the exact origin of the slab component is not the topic of this work and a discussion of the possible formation of the slab component is beyond the purpose of this manuscript. Thus, we do not refer to specific papers on different slab components.**

L172: You jumped from Fig. 1 in previous sections to Fig. 8, which is missing I can't find it!!!. **Sorry, the Ar-Ar figure is now in the supplement.**

L182: one sentence is needed to summarize this new age data. **Added**

L198: This is not clear, there is no difference between Nd values between old and young Futuna Flank data? And the same for Ce/Pb ratio and even the Tanna samples?? **This sentence was indeed confusing and we have rewritten it and refer more to figures. We hope our statements are clearer now.**

L272: Some other ratios Ba/Nb, U/Nb, Pb/Nd, can help in your discussion and show the differences in slab contributions between the two types, please see Gamaleldien et al, 2020, SR (start of subduction paper). I like to see these ratios vs. Pb, Nd, Sr isotopes. These figures can be added to supplementary. **We added a figure with Ba/Nb and one with additional incompatible element ratios (new Fig. 4), but U/Nb is quite scattered, probably due to alteration, and we show a figure with Ce/Pb (Fig. 5b) which is very similar to Pb/Nd. Thus, we do not think that more figures help to make our points and the reviewers were satisfied with the number of figures and thought they were convincing.**

Fig. 1: Please add a more general map to Fig. 1 to show the location for the general/public readers. The symbol for Tanna Yasur "red triangle" can't be found in the following figures!!! **We have redrawn Figure 1 and we hope that locations and symbols can be observed better now.**

Fig. 2: Can you add the age for Tanna's old and young lava ?? **OK, there is not much radiometric age data, but we added the known ages.**

Fig 5: Can you add Ba/La vs Pb isotopic data? **We added Ba/Nb versus $^{206}\text{Pb}/^{204}\text{Pb}$ because Ba/Nb shows distinct compositions for the subduction-related lavas and enriched MORB.**

Fig 6: Please explain how you did the mixing model? **The end-member compositions of the**

mixing model are now explained in the figure caption.

Fig. 7: please label the different components of the figure such as oceanic crust, mantle wedge, I think this figure can be more visualized than the present one?? **We tried to use the three components from the previous figures, i.e. Depleted Mantle, Enriched Mantle, Slab Component in order to be consistent. Additional labels would make the figure to complicated.**

Reviewer #3 (Remarks to the Author):

This is an interesting, valuable paper. The different lines of evidence the authors use to build their story – rock ages, geochemistry, location, tectonic history – are effectively described and synthesized. The tectonic setting, strategy and conclusions are unique, so the paper is definitely appropriate for Nature Communications.

Nonetheless, the paper can be improved for greater clarity and impact.

Clarity - To be fully understood and effective, the paper needs some work. In a short work, readers will compare text and figures as they read, so the the figures should to be explicit and consistent.

1) The most important item is Fig. 1. The lettering in 1a is tiny (to my eyes). Every feature mentioned in the Results and especially Discussion (Evolution of the tectonic and magmatic processes) should be labelled so that readers can correlate the upper plate locations with interpretations of the sample chemistry (Geochemical evolution of the magmas) and processes occurring at depth (Implications of the geochemical evolution for tectonic processes). A box showing the location of 1b on Fig. 1a would be useful. **We have redrawn Figure 1 and hope it is better now.**

2) Also, in general, the figure captions are brief and should be expanded to cover everything on the diagram. Some examples – no explanation the different symbols on Fig. 1b, no explanation of the yellow field on Fig. 4, samples identified only as numbers on Fig. 3 legend (these are in the caption but are better on both), explanation of open green circles on Fig. 6. **Extended and we hope that the figures are better explained now.**

3) Supplementary data: Some small editing needed – Supplementary Data 1 has all geochemical data for samples (Table S1) and standards (Standards Tab). Supplementary Data 2 has Ar dates and analytical parameters in Tabelle 1 and nothing in Tabelle 2. **Changed**

Broadening the impact:

The authors emphasize that the mantle flow rate they infer is fast (line 55), so some comparisons with rates of mantle flow inferred for other arcs is needed. Confirmation of results with those of with Heuret and Lallemand is good, but comparison with mantle wedge flow rates estimated in the literature elsewhere would heighten the impact of the paper and enhance the value of the authors' conclusions. **We tried to expand this discussion and hope it is better now.**

I've attached an annotated version of the manuscript with further comments.

Line 31: Replaced tectonically by seismically, and hope this makes more sense

Line 41: The corner flow is believed to be induced by the subduction of the slab and appears to be in agreement with many models and observations although 3D models often indicate flow parallel to the arc (see e.g. Wiens et al 2009 Ann Rev Earth Planet Sci).

Line 168: We refrained of putting the entire methods section into the supplement because we believe it contains important information and is not too long.

Line 172: Second "in" deleted

Line 193: Added to Figure 3.

Line 222: Should be all in Figure 1 now.

Line 271: Replaced

Line 277: We agree this may be possible but the fact that the mixing trends imply two very distinct mantle compositions that do not show mixing like it is observed in the North Fiji Basin basalts implies to us that there are indeed two separate mantle portions in the melting region of the back-arc that changed at about 2.5 Ma.

Line 280: Because the mantle end-members resemble North Fiji Basin basalts we believed that both the enriched and the depleted mantle originated further to the east beneath the North Fiji Basin. Due to the flow induced by the slab at the New Hebrides arc the enriched mantle was replaced by depleted mantle after 2.5 Ma. We hope this is now clearer in the text and Figure 8.

Line 287: The mixing model is now explained in the caption of Figure 7.

Line 303: Thank you

Line 317: We hope we explained our model better in the text and in Figure 8.

Line 343: We explain this in more detail and we think that these are different mantle portions flowing in from the east rather than depletion process that would probably not jump from an enriched to a depleted end-member but would sample compositions in between.

Line 348: We added some sentences to discuss this issue in more detail.

Line 551: Ok, added

Line 556: Explained in figure now

Line 564: They are more MORB-like than our samples and we speculate this is because they are younger and closer to the most recent rift, but we cannot prove that. Thus, we did not discuss this issue in great detail but added a sentence to the caption.

REVIEWER COMMENTS

Reviewer #1 (Remarks to the Author):

1. I wonder if there is any other evidence supporting the slab steepening process. In the Izu- Bonin-Marina subduction zone, slab steepening was suggested by the gradual narrowing area of the arc magmatism (Ishizuka et al., 2009) and the migration of the rear-arc volcanism towards the trench (Miyazaki et al., 2020). The more slab melt component found in the rear-arc basalts of the Izu-Bonin arc also reveals a deep mantle condition at which the temperature is high enough to induce slab melt (Miyazaki et al., 2020). I think it is necessary to find more evidence supporting your argument that the New Hebrides slab may have steepened since 2.5 Ma. Presently, the New Hebrides slab is one of the steepest on Earth and the opening of the Coriolis Troughs indicates that the upper plate stress field became extensional some 2.5 Ma ago. We believe that the most likely explanation is the steepening of the plate which is supported by numerical and experimental models e.g. by Cagnioncle et al (2007, JGR) and Grove et al (2012, Nature).

I do not think that the authors answer my comment in a straightforward and explicit way. My comment is more observations would be good to support slab steepening. I am afraid my comment is not answered by simply referring to these two papers.

2. Slab-deepening process in the Izu-Bonin subduction zone leads to a series of reararc volcanisms featured by increasingly enriched features with time (Miyazaki et al., 2020). This is contrary to the arc geochemistry variation shown in this study which is characterized by the increase in depletion. I think a comparison could be made between the two subduction zones to find the key process controlling the differences, which may better explain your ideas. We believe that the steep slab allows rapid inflow of unmetasomatized mantle which may not be the case in the Izu-Bonin. MORB-like mantle is also observed in back-arcs of other subduction zones with steep slabs like Tonga, South Sandwich and the Marianas and we have

mentioned this in the discussion.

I am not convinced by this (over) concise reply.

4. The New Hebrides subduction evolved for quite a while (>10 Myr), and the subducted slab is quite deep, reaching the mantle transition zone. Why slab steepening occurred at this particular moment (i.e., 2.5 Ma)? We believe that the slab rollback has probably occurred in several phases with slab steepening and the event at 2.5 Ma was only the last in a series of such events but this is of course highly speculative.

My comment is not answered explicitly.

- Line 185: It's good to mark the absolute age by linear color bar in Fig.2. Same for the Figs. 4, 5 and 6. We are not sure whether we understand this comment but we have redrawn many of the figures.

What I mean is that the ages of the samples/symbols could be indicated by colors by showing a colorbar.

Reviewer #2 (Remarks to the Author):

Thanks for addressing most of my comments on the previous version. The authors did a good job of making the manuscript clear and ready for publication.

Please note you mentioned in the manuscript Fig.S3 and I can't find Fig.S1 and S2?!

Reviewer #3 (Remarks to the Author):

The authors have addressed all of my comments. The revised version, including text and figures, is clear and understandable. A short discussion of mantle flow rates has been added to the last discussion section. I recommend acceptance for publication in Nature Communications.

Replies to reviewers

Reviewer 1

1. I wonder if there is any other evidence supporting the slab steepening process. In the Izu-Bonin-Marina subduction zone, slab steepening was suggested by the gradual narrowing area of the arc magmatism (Ishizuka et al., 2009) and the migration of the rear-arc volcanism towards the trench (Miyazaki et al., 2020). The more slab melt component found in the rear-arc basalts of the Izu-Bonin arc also reveals a deep mantle condition at which the temperature is high enough to induce slab melt (Miyazaki et al., 2020). I think it is necessary to find more evidence supporting your argument that the New Hebrides slab may have steepened since 2.5 Ma.

Presently, the New Hebrides slab is one of the steepest on Earth and the opening of the Coriolis Troughs indicates that the upper plate stress field became extensional some 2.5 Ma ago. We believe that the most likely explanation is the steepening of the plate which is supported by numerical and experimental models e.g. by Cagnioncle et al (2007, JGR) and Grove et al (2012, Nature).

I do not think that the authors answer my comment in a straightforward and explicit way. My comment is more observations would be good to support slab steepening. I am afraid my comment is not answered by simply referring to these two papers.

We do not think that the Izu-Bonin subduction zone is different from the New Hebrides, at least if the last 8 Ma are studied. During the past 8 Ma the volcanism of the Izu-Bonin back-arc migrated towards the arc front (Ishizuka et al 1998 Island Arc, Ishizuka et al 2003 J Volcanol Geotherm Res) and the youngest volcanism (< 1 Ma) appears to be restricted to the rift zones of the Sumisu and Torishima Basins and smaller basins to the north (Hofstaedter et al 1990 EPSL). The lavas in these basins have less slab input than those from the arc front and than the older lavas of the back-arc knolls (e.g. rift lavas have mostly $^{143}\text{Nd}/^{144}\text{Nd} > 0.51305$, whereas back-arc knolls have lower $^{143}\text{Nd}/^{144}\text{Nd}$, Tollstrup et al 2010 G cubed). Thus, whereas the 10 to 2 Ma old back-arc knoll lavas often contain a significant amount of deep sediment melt, the young rift lavas do not show a strong sediment component. This was noted by Ishizuka et al (2006, J Volcanol Geotherm Res) and explained to reflect a change from sediment melt in the back-arc knoll lavas to slab fluid in the young rift lavas. In fact, these authors also suggest that slab steepening could explain the change similar to our model for the New Hebrides. Thus, we thank the reviewer to suggest to look closer at the Izu-Bonin back-arc and we believe that the observations in the Sumisu Rift closely resemble those in the Futuna Trough. We have added a discussion to the manuscript (Lines 347-359) to explain the similarities and suggested model of slab steepening by Ishizuka et al (2006, J Volcanol Geotherm Res).

2. Slab-deepening process in the Izu-Bonin subduction zone leads to a series of reararc volcanisms featured by increasingly enriched features with time (Miyazaki et al., 2020). This is contrary to the arc geochemistry variation shown in this study which is characterized by the increase in depletion. I think a comparison could be made between the two subduction zones to find the key process controlling the differences, which may better explain your ideas.

We believe that the steep slab allows rapid inflow of unmetasomatized mantle which may not be the case in the Izu-Bonin. MORB-like mantle is also observed in back-arcs of other subduction zones with steep slabs like Tonga, South Sandwich and the Marianas and we have mentioned this in the discussion.

I am not convinced by this (over) concise reply.

We looked more into the evolution of Izu-Bonin in the last 8 Ma and found that there is a striking similarity in the narrowing of the back-arc volcanic region, the migration of the volcanism towards an active rift zone next to the arc front, and a decrease of the slab input into the back-arc magmas (Ishizuka et al, 2003, 2006 J Volcanol Geotherm Res). The lavas in the Sumisu and Torishima Rifts are unlike the arc front and older back-arc lavas (Hofstaedter et al 1990 EPSL, Tollstrup et al 2010 G cubed, Fryer et al 1990 EPSL) and thus resemble the Futuna Trough lavas. We discuss this in line 347-359.

4. The New Hebrides subduction evolved for quite a while (>10 Myr), and the subducted slab is quite deep, reaching the mantle transition zone. Why slab steepening occurred at this particular moment (i.e., 2.5 Ma)?

We believe that the slab rollback has probably occurred in several phases with slab steepening and the event at 2.5 Ma was only the last in a series of such events but this is of course highly speculative.

My comment is not answered explicitly.

The rifting occurred at the same time assumed for the collision between the D'Entrecasteaux Ridge and the New Hebrides arc and models suggest that this caused a more rapid clockwise rotation of the slab with a steepening of the shallow subduction dip. We have added two sentences on this matter at line 405 ff.

• Line 185: It's good to mark the absolute age by linear color bar in Fig.2. Same for the Figs. 4, 5 and 6.

We are not sure whether we understand this comment but we have redrawn many of the figures.

What I mean is that the ages of the samples/symbols could be indicated by colors by showing a colorbar.

We have used different colours for the different settings and believe that adding more colours for different ages may be too confusing. We hope that the age variations become clear in the figures.

Reviewer #2 (Remarks to the Author):

Thanks for addressing most of my comments on the previous version. the authors did a good job of making the manuscript clear and ready for publication.

Please note you mentioned in the manuscript Fig.S3 and I can't find Fig.S1 and S2?!

We have three Supplements and S3 is a Figure. We hope that this is OK.

REVIEWERS' COMMENTS

Reviewer #1 (Remarks to the Author):

I am pleased to see that all my comments are thoroughly addressed. I do not have any further comments.